

**Anthropogenic influences on the physical state of submicron particulate**
**matter over a tropical forest**
Adam P. Bateman[1], Zhaoheng Gong[1], Tristan H. Harder[2], Suzane S. de Sá[1], Bingbing Wang[3,4],
Paulo Castillo[5], Swarup China[3], Yingjun Liu[1], Rachel E. O'Brien[2,6], Brett Palm[7], Hung-Wei
Shiu[3], Glauber da Silva[8], Ryan Thalman[5], Kouji Adachi[9], M. Lizabeth Alexander[3], Paulo
Artaxo[10], Allan K. Bertram[11], Peter R. Buseck[12], Mary K. Gilles[2], Jose L. Jimenez[7], Alexander
Laskin[3], Antonio O. Manzi[8], Arthur Sedlacek[5], Rodrigo A. F. Souza[13], Jian Wang[5], Rahul
Zaveri[3], Scot T. Martin[*1,14]
[1]School of Engineering and Applied Sciences, Harvard University, Cambridge, Massachusetts,
USA
[2]Chemical Sciences Division, Lawrence Berkeley National Laboratory, Berkeley, CA, USA
[3]William R. Wiley Environmental Molecular Sciences Laboratory, Pacific Northwest National
Laboratory, Richland, WA, USA
[4]State Key Laboratory of Marine Environmental Science, College of Ocean and Earth Sciences,
Xiamen University, Xiamen, China
[5]Brookhaven National Laboratory, Upton, New York, USA
[6]Department of Civil and Environmental Engineering, Massachusetts Institute of Technology,
Cambridge, Massachusetts, USA
[7]University of Colorado, Boulder, USA
[8]National Institute of Amazonian Research, Amazonas, Brazil
[9]Atmospheric Environment and Applied Meteorology Research Department, Meteorological
Research Institute, Tsukuba, Ibaraki, Japan
[10]University of São Paulo, São Paulo, Brazil
[11]Department of Chemistry, University of British Columbia, Vancouver, BC, Canada
[12]School of Earth and Space Exploration & School of Molecular Sciences, Arizona State
University, Tempe, Arizona, USA
[13]Amazonas State University, Amazonas, Brazil
[14]Department of Earth and Planetary Sciences, Harvard University, Cambridge, Massachusetts,
USA
*Atmospheric Chemistry and Physics*
[*] Correspondence to: S.T. Martin (scot_martin@harvard.edu)





**Abstract**
The occurrence of non-liquid and liquid physical states of submicron atmospheric
particulate matter (PM) downwind of an urban region in central Amazonia was investigated.
Measurements were conducted during two Intensive Operating Periods (IOP1 and IOP2) that
took place during the wet and dry seasons, respectively, of the GoAmazon2014/5 campaign. Air
masses representing variable influences of background conditions, urban pollution, and regional
and continental scale biomass burning passed over the research site. As the air masses varied,
particle rebound fraction, which is an indicator of the mix of physical states in a sampled particle
population, was measured in real time at ground level using an impactor apparatus. Micrographs
collected by transmission electron microscopy confirmed that liquid particles adhered while non-
liquid particles rebounded. Relative humidity (RH) was scanned to collect rebound curves. When
the apparatus RH matched ambient RH, 95% of the particles were liquid as a campaign average,
although this percentage dropped to as low as 60% during periods of anthropogenic influence.
Secondary organic material, produced for the most part by the oxidation of volatile organic
compounds emitted from the forest, was the largest source of liquid PM. Analyses of the mass
spectra of the atmospheric PM by positive-matrix factorization (PMF) and of concentrations of
carbon monoxide, total particle number, and oxides of nitrogen were used to identify time
periods affected by anthropogenic influences, including both urban pollution and biomass
burning. The occurrence of non-liquid PM correlated with these indicators of anthropogenic
influence. A linear model having as output the rebound fraction and as input the PMF factor
loadings explained up to 70% of the variance in the observed rebound fractions. Anthropogenic
influences appear to favor non-liquid PM by providing molecular species that increase viscosity



when internally mixed with background PM, by contributing non-liquid particles in external
mixtures of PM, and a by combination of these effects under real-world conditions.




## 1. Introduction


Particulate matter (PM) directly affects the Earth's climate by scattering and absorbing

solar radiation and indirectly by effects on clouds (Ramanathan et al., 2001). The magnitude of
these effects depends in part on the physical and chemical properties of the particulate matter
(Andreae and Rosenfeld, 2008). The physical state of PM, as liquid or non-liquid, can influence
the growth rates of small particles and ultimately the production of cloud condensation nuclei
(CCN) (Riipinen et al., 2011;Perraud et al., 2012). Liquid particles pose negligible in-particle
diffusion barriers for condensing species and therefore can grow rapidly. By comparison, non-
liquid particles, referring to both semisolid and solid particles, can have a different behavior. For
some conditions, semisolid particles can grow slowly because of in-particle limits on rates of
molecular diffusion, and solid particles can grow even more slowly when limited to surface
adsorption (Riipinen et al., 2012;Shiraiwa and Seinfeld, 2012;Li et al., 2015). Liquid compared
to non-liquid PM can also affect reactivity (Kuwata and Martin, 2012;Li et al., 2015). The
consequences of the differing growth mechanisms can be that the growth of small particles is
relatively disfavored in a population of liquid particles of heterogeneous sizes, as compared to a
similar population of non-liquid particles (Zaveri et al., 2014). An implication can be that CCN
concentrations are ultimately greater for a population of non-liquid particles that grows to CCN
sizes, as compared to a population of liquid particles.

Secondary organic material (SOM), produced by the oxidation of biogenic volatile

organic compounds (BVOCs), is a major source of atmospheric PM, especially over forested
regions where SOM often dominates the mass concentration of submicron PM (Hallquist et al.,
2009;Jimenez et al., 2009). The physical state of SOM has been studied in both laboratory
(Vaden et al., 2011;Kuwata and Martin, 2012;Perraud et al., 2012;Saukko et al., 2012;Renbaum-





Wolff et al., 2013;Kidd et al., 2014;Bateman et al., 2015;Li et al., 2015;Liu et al., 2015;Pajunoja
et al., 2015;Song et al., 2015) and ambient environments (Virtanen et al., 2010;O'Brien et al.,
2014;Bateman et al., 2016;Pajunoja et al., 2016). For background conditions of the Amazonian
tropical forest, a region dominated by isoprene-derived SOM and high RH, PM was mostly
liquid (Bateman et al., 2016). For a boreal forest in northern Europe, a region dominated by
pinene-derived SOM and low RH, PM was largely non-liquid (Virtanen et al., 2010). The
combined set of laboratory and ambient studies show that the physical state of PM having high
SOM content depends on the surrounding relative humidity (RH). This effect arises in part
because organic particles are hygroscopic to various extents depending on composition. Water
absorption, which is favored at elevated RH, has a plasticizing effect on physical state (Koop et
al., 2011).

The physical state of PM affected by urban pollution over forests remains largely

unexplored. A single day of ambient observations in central Amazonia suggested that ambient
PM affected by urban pollution tended toward a non-liquid state (Bateman et al., 2016), and
laboratory studies support the idea of an important modulating role of pollution in the physical
state of SOM. A non-liquid state was favored for SOM produced from single-ring aromatic
species (Liu et al., 2015;Song et al., 2015) as well as mixed with polycyclic aromatic
hydrocarbons (PAHs) (Zelenyuk et al., 2012;Abramson et al., 2013). Organic molecules
associated with urban pollution and industrial activities tend to be less hygroscopic than biogenic
SOM (Hersey et al., 2013). When internally mixed within biogenic SOM, the anthropogenic
molecules have a tendency to reduce water uptake and thereby reduce the viscosity of the mixed
particles. A similar line of reasoning leads to an identical hypothesis for the effects of biomass
burning emissions. Compared to prevailing background conditions of SOM dominance over



many forests, PM produced by biomass burning leads to a net effect of decreased water uptake
when the PM mixes into the background particle population (Dusek et al., 2011).

The data sets presented herein provide observational evidence on the effects of

anthropogenic influences on the physical state of ambient particulate matter. All other factors
being equal, the lack of particle rebound is an indicator of liquid PM (Bateman et al., 2015).
Conversely, the occurrence of particle rebound is an indicator of non-liquid PM. Rebounding and
adhering particles were separately collected for conventional and chemical imaging. The data
sets were collected during the two Intensive Operating Periods (IOP1 and IOP2) of the
GoAmazon2014/5 experiment, corresponding to the wet and dry seasons, respectively (Martin et
al., 2016). The research site (-3.2133°, -60.5987°), called "T3", was located 70 km downwind of
the city of Manaus, population two million, in central Amazonia. Air masses representing
background conditions, urban pollution, and regional and continental scale biomass burning
passed over the research site. Herein, anthropogenic influence refers to all but background
conditions. Under background conditions, the submicron PM in this region is dominated by
biogenic SOM (Chen et al., 2009;Chen et al., 2015).
**2. Experimental**

An impactor apparatus was used for the study of particle rebound (Bateman et al.,

2014;Bateman et al., 2015;Bateman et al., 2016). The apparatus was housed inside a
temperature-controlled research trailer at the T3 site. Particulate matter was sampled at 5 m
above ground through copper tubing having an outer diameter of 13 mm (0.5 inch). In sequence,
a drying unit reduced the sampled flow to 25% RH or lower, a Differential Mobility Analyzer
(DMA, TSI 3085) selected a subpopulation of dried particles having a narrow distribution of
electric mobility, and a humidification unit (Nafion tubes; Perma Pure, MD 110) elevated the RH



of the mobility-filtered flow to the targeted RH of a measurement. The drying unit consisted of a
Nafion drier in series with a silica gel diffusion drier, and the silica gel was replaced every two
days. After passing the dryer, DMA, and humidifier, the resulting flow was split and passed
through three impactors operated in parallel. Labels *i*, *ii*, and *iii* refer to each of the three
impactors. Each impactor was operated at a flow rate of 1.0 Lpm, corresponding to a setpoint
aerodynamic diameter $d_a^*$ of 84.9 ± 5.4 nm (Bateman et al., 2014). Particle number
concentrations, denoted by $N_i$, $N_{ii}$, or $N_{iii}$, exiting the impactors were measured by three
independent condensation particle counters (CPC, TSI 3010). Measurements were conducted
from February 14 to March 16, 2014, during IOP1 and from September 4 to October 15, 2014,
during IOP2.

The three impactors differed from one another by having uncoated, coated, or no

impaction plate. The impactor having the uncoated plate passed both non-impacted and
rebounded particles. The impactor having the coated plate (Dow Corning High-Vacuum Grease)
passed only non-impacting particles. The impactor having no plate passed all particles. Its
purpose was to serve as a compensation arm for possible miscellaneous particle losses, such as
wall loss. Based on the particle number concentrations measured downstream of the impactors,
the rebound fraction was calculated as follows (Bateman et al., 2014):
$$f = \frac{N_i - N_{ii}}{N_{iii} - N_{ii}} \tag{1}$$
The terms $N_i$, $N_{ii}$, and $N_{iii}$ represent the particle number concentration measured downstream of
the impactors having uncoated, coated, and no impaction plate, respectively. The standard
deviation of the rebound fraction was based on error propagation for an uncertainty in $N$ of $N^{1/2}$
(Agarwal and Sem, 1980).



A rebound curve representing $f$(RH) constituted an individual data set. For most

measurements, the DMA setting for electric mobility was held constant (typically 190 nm), the

RH in the humidification unit was changed stepwise every few minutes, and $f$ was continuously

recorded. Additional protocols of DMA settings and RH profiles were used in a few cases. All

protocols, including differences between IOP1 and IOP2, are described in Section S1 of the

Supplement.

In conjunction with the rebound measurements, particles were collected for imaging by

transmission electron microscopy (TEM) and scanning transmission X-ray microscopy (STXM).

Images in some cases can directly suggest the liquid or non-liquid state of individual particles

(O'Brien et al., 2014;Wang et al., 2016). Samples for imaging were collected as follows. A fourth

impactor was added in parallel with the other three impactors. Imaging substrates were affixed to

the impaction plate, and multiple substrates were mounted to the plate for concurrent collection

for the various microscopy techniques. Substrates included grids coated with Formvar (EMJapan

Co., Tokyo, Japan) or lacey carbon (Ted Pella Inc., USA) for TEM and silicon nitride membrane

windows (Silson, UK) for STXM. The flow rate and setpoint aerodynamic diameter of the fourth

impactor were the same as for the other three impactors. The collected particles represented

those that adhered to the substrate at impact, and an assumption in the analysis is that rebound

from the substrate was similar to that from the uncoated plate. The flow through this fourth

impactor was pulled by an in-line TEM autosampler (Arios Inc., Tokyo, Japan (Adachi et al.,

2014)). In this way, the setup separately collected particles that adhered to the grids on the

impaction plate and particles that rebounded from the impaction plate and passed to the

autosampler. The autosampler collected particles having aerodynamic diameters from 60 to 350

176       nm. Particles adhered to the TEM substrates in the autosampler yet rebounded from the TEM





substrates in the impactor because of the significantly different particle impact velocities
between the two impactors (i.e., cut-point and impactor design). Samples were collected for
TEM analysis between September 30 and October 15, 2014. Samples were collected for STXM
analysis between 1:00 and 10:00 (UTC) on October 1, 2014.

Microanalysis of individual particles of the collected PM was performed using two

instruments: (1) a transmission electron microscope (TEM; JEOL, JEM-1400) equipped with an
energy-dispersive X-ray spectrometer (EDS; Oxford Instruments) and (2) a scanning
transmission X-ray microscope interfaced for near-edge X-ray absorption fine structure
spectroscopy (STXM/NEXAFS; Advanced Light Source, Berkeley). For TEM, imaging was by
bright-field microscopy, and particle composition was investigated by EDS. For STXM, at a
fixed photon energy an image was obtained by detecting the transmitted light at each pixel while
raster scanning the sample. Spatially resolved NEXAFS spectra were obtained from a set of
images recorded at different photon energies. The NEXAFS spectra provided chemical bonding
information and quantitative elemental ratios (Moffet et al., 2010a;Moffet et al., 2010b;O'Brien
et al., 2014;Piens et al., 2016). Section S2 of the Supplement presents further technical
information concerning the TEM and STXM/NEXAFS analyses.

Additional co-located measurements used in the data analysis herein included a High-

Resolution Time-of-Flight Aerosol Mass Spectrometer (AMS; Aerodyne Inc.) (Hu et al., 2015;de
Sá, 2016;Hu et al., 2016), a Single-Particle Soot Photometer (SP2; Droplet Measurement
Technologies), a size-resolved Cloud Condensation Nuclei Counter (CCNC; Droplet
Measurement Technologies, CCNC-100) (Thalman, 2016), a Condensation Particle Counter
(TSI, CPC 3772) for measuring particle number concentrations, an Integrated Cavity Output
Spectroscope (ICOS; Los Gatos) for measuring carbon monoxide (CO) concentration, and a



Trace Level Enhanced detector (TLE; Thermo Scientific, Model 42i, with further customization)
for the measuring  concentrations of nitrogen oxides ($NO_y$). The latter three instruments were
deployed at the T3 site as part of the USA Department of Energy (DOE) Atmospheric Radiation
Measurement (ARM) Climate Research Facility, including the ARM Mobile Facility One (AMF-
1) and the Mobile Aerosol Observing System (MAOS) (Mather and Voyles, 2013;Martin et al.,
2016). This facility also collected the meteorological data used herein, including temperature,
relative humidity, wind direction, and wind speed. The AMS measured the chemical composition
in real-time of non-refractory submicron particles (DeCarlo et al., 2006). The high-resolution
data of the organic component in "V-mode" was used in conjunction with positive-matrix
factorization (PMF) analysis (Ulbrich et al., 2009) to obtain statistical factors and the associated
time series of factor loadings for each season (de Sá, 2016). As a surrogate for concentrations of
black carbon, the SP2 measured the time-resolved scattering and incandescence produced by
irradiated refractory, light-absorbing components of individual particles having volume-
equivalent diameters between 70 and 600 nm (Schwarz et al., 2006;Moteki and Kondo, 2007).
The size-resolved CCN activity was obtained by classifying dry particles using a DMA (TSI
3081) and then exposing them successively to a range of supersaturations inside the CCNC
(Thalman, 2016). The hygroscopicity and CCN activity were analyzed via κ-Köhler theory,
which relates particle critical supersaturation to the initial dry diameter and hygroscopicity
(Petters and Kreidenweis, 2007;Petters et al., 2009).
**3. Results and Discussion**
3.1 *Rebound Observations*

Particle rebound or particle adhesion at impact depends on the balance of energies.

Particle rebound occurs when the kinetic energy before impact is greater than the sum of



dissipation and surface adhesion energies after impact (Tsai et al., 1990;Bateman et al., 2014).
The dissipation energy of liquid particles is much greater than that of solid particles because of
additional mechanisms of dissipation available to the former. Calibration of the impactor shows a
transition from rebound to adhesion between $10^2$ to 1 Pa s in viscosity for sucrose particles
(Bateman et al., 2015).

The rebound curves of particles of 190-nm mobility diameter are shown in Figure 1a for

IOP1 and IOP2. For RH < 50%, the rebound fraction was between 0.8 and 1.0. For RH > 50%,
the rebound fraction decreased monotonically to a low value, typically zero. The shape of the
rebound curve in Figure 1a is similar to that for particles of secondary organic material produced
in the Harvard Environmental Chamber (Bateman et al., 2015). The rebound fraction of SOM
particles produced from isoprene or α-pinene became zero for RH > 90%.

The subset of the data for which the apparatus RH matched the ambient RH is shown in

Figure 1b  (cf. Section S3 of Supplement). In light of the distribution of ambient RH values (cf.
Figure 1c), the data set in Figure 1b implies that submicron PM in this tropical environment was
liquid most of the time. Bateman et al. (2016) reported such a result for observations under
background conditions at this site for data sets collected in 2013 a few months before
GoAmazon2014/5 began. The present results reported for GoAmazon2014/5 represent a longer
time series (i.e., 550 compared to 30 rebound curves) and  reinforce the generality of the earlier
result of Bateman et al. (2016) of the prevalence of liquid PM for this forested region under
background conditions.

In the larger set of observations of the present study, what also emerges is that the

rebound fraction remained in median at 0.05 even at 95% RH, implicating the presence of
externally mixed PM in the atmosphere. Approximately 5% of the particles were non-liquid even



to 95% RH in both the wet and dry seasons (i.e., IOP1 and IOP2) (Figures 1a and 1b). At times,
rebound fractions of up to 0.4 occurred for RH > 90%. Elevated rebound fractions were observed
more frequently in IOP2 than IOP1. During IOP2, which was extensively influenced by biomass
burning, non-liquid particles in median constituted a fraction of 0.3 at 70% RH and a fraction of
0.1 at 90%. These RH values prevailed approximately 10% of the time. Events of elevated
rebound fraction were associated with the pollution plume from Manaus during IOP1 in the wet
season and with both this urban plume as well as increased regional biomass burning during
IOP2 in the dry season (cf. Section 3.2).

Probability density functions (PDF) of rebound fraction at 75% RH based on the data

sets of Figure 1a are shown in Figure 2 for (a) IOP1 and (b) IOP2 for three types of air masses
during day and night time periods. Daytime represents 12:00 to 16:00 (local time) (16:00 to
20:00 UTC) and nighttime represents 23:00 to 04:00 (local time) (03:00 to 08:00 UTC). Air
masses were identified as background, as influenced by local to regional biomass burning, or as
influenced by Manaus pollution. The classification scheme was based on concentration regimes
of particle number, carbon monoxide, and odd nitrogen ($NO_y$), as presented in Section S4 of the
Supplement. Background conditions included contributions both from natural processes and
from the long-distance transport and extensive oxidation of biomass burning emissions (Chen et
al., 2009;Martin et al., 2010). Natural processes in this region were dominated by the production
of secondary organic material from oxidation of plant emissions (Martin et al., 2010;Poschl et
al., 2010). Figure 2a shows that the mode value of the PDF shifted higher under the influence of
Manaus pollution compared to background conditions, meaning that rebound became more
probable and indicating an increasing presence of non-liquid PM at 75% RH. In Figure 2b, the
mode value shifted even higher under the influence of biomass burning. For each type of air





mass, the nighttime mode values were higher than the daytime equivalents. The day-night
differences were lowest under background conditions.
The increase in rebound at night might be explained by a combination of interacting
factors. A shallow stable nocturnal boundary layer can trap and thereby concentrate
anthropogenic non-liquid particles emitted at night from local emissions, including smoldering
fires during IOP2. During the day, the boundary layer expands and dilution is more effective. In
addition, the production of liquid secondary organic material by biogenic processes is
comparatively more rapid.
For further analysis, the rebound curves recorded under background conditions were
averaged separately for IOP1 and IOP2 (cf. Figure 1a and Section S5 of the Supplement). These
two background-average curves served as references against which deviations in rebound
fraction were calculated for all air masses. Rebound deviation represents the excess non-liquid
PM over background conditions after detrending the data for the dependence on relative
humidity. The rebound deviations for IOP1 and IOP2 are plotted in Figure 3. Rebound deviations
as high as +0.5 in rebound fraction were observed. Statistics of Figure 3 are summarized in Table
S1, including for the subset of measurements matched with the ambient RH. For the full set of
data, including the full range of RH, rebound deviations greater than 0.1 represented 17% and
35% of the observations during IOP1 and IOP2, respectively. These deviations, corresponding to
increased rebound and thus indicating an increasing presence of non-liquid PM, corresponded to
the anthropogenic influences, as developed further in Section 3.2.





3.2 *Relationships between Rebound and Other Observations*
3.2.1 Case Studies

Transitions between background and polluted conditions across 24 hour-periods are

presented in Figure 4 as representative examples for each IOP. The bottom panel shows the
deviation in rebound fraction relative to the background-average curve (cf. Section S5 of
Supplement). Color coding distinguishes relative humidity. From bottom to top, other panels in
the figure show temperature, wind direction, wind speed, relative fractions of two groups of
AMS PMF factor loadings, black carbon (BC) concentration, sulfate concentration, and total
submicron PM mass concentration. Processes contributing to the loading of PMF group A are
largely associated with the background atmosphere of Amazonia, including the possibility of
long-range transport and extensive oxidation of biomass burning emissions. Processes
contributing to the loading of PMF group B are largely associated with urban pollution and local
and regional biomass burning (cf. Section S6 of Supplement).

Shifts in the deviation in the rebound fraction from the background-average curve are

apparent in the time series in the bottom panels of Figure 4. At these times, the fractional loading
of PMF group A decreased and that of PMF group B increased, indicating a shift away from
background conditions. Background conditions were characterized by high loadings of PMF
group A and small rebound deviations. In the left panel (IOP1), in an example of one type of an
anthropogenic event, black carbon concentration and the fractional loading of PMF group B
abruptly increased together. The rebound deviation simultaneously increased, indicating an
increasing presence of non-liquid PM, especially above 70% RH. In the right panel (IOP2), in an
example of a second type of an anthropogenic event, the background air mass was gently
replaced by an air mass characterized by small increases in rebound deviation and fractional



loading of PMF group B yet lacking an associated increase in black carbon concentration. In an
example of a third type of anthropogenic event, this air mass was later replaced by an air mass
characterized by a strong increase in the fractional loading of PMF group B, total PM mass
concentration, and rebound deviation. Around 15:00 (UTC), a convective event associated with
rainfall decreased temperature, changed wind direction, and increased wind velocity, and
background conditions returned.

TEM images collected during the time periods of elevated rebound corroborate the

foregoing interpretation of liquid and non-liquid PM for adhering compared to rebounding
particles. Images of particles that adhered to the TEM substrates in the impactor at 95% RH are
shown in Figure 5a. The images were taken at a tilt angle of 60° so that the aspect ratio of the
particles could be viewed. The images show that most adhering particles spread out across the
substrate, indicating flattening upon impact, as expected for liquid particles. Some of these
adhering particles had a mixed composition, appearing as solid cores surrounded by halos of
flattened liquid shells. The horizontal dimensions of the flattened particles approached 1 to 2 μm
for vertical dimensions of tens of nanometers. Pöschl et al. (2010) previously recorded similar
images for ambient particles in the wet season of 2008 in central Amazonia during the
Amazonian Characterization Experiment (AMAZE-08) (Martin et al., 2010). For comparison to
the images of the adhering particles, images of particles that rebounded at 95% RH from the
impaction plate are shown in Figure 5b. These particles were collected downstream of the
impactor. The images show that these particles had high vertical dimensions and spherical or
dome-like morphologies and thus suggest that the particles experienced little deformation upon
impact, as expected for solid particles.




3.2.2 Descriptive Statistics

The results of Figures 3 and 4 are further analyzed in Figure 6. Statistics of rebound

deviation are shown by box-whisker representation for different windows of relative humidity.
The data sets were segregated for presentation by IOP1/IOP2, daytime/nighttime, and air mass
type. For background conditions, the rebound deviations relative to their average were mostly
zero, indicating that there was low variability among different background air masses. The
exception was for the night periods of IOP1. By comparison, under anthropogenic influences, the
rebound deviation was positive for both IOPs. Positive deviations were most significant between
65% and 95% RH. In all cases, the nighttime deviations were greater than the daytime
counterparts. For IOP2, the prevalence of biomass burning confounded separate classifications of
urban pollution and biomass burning, and a classification of biomass burning took precedent.
Rebound deviations were strongest during these time periods. Statistics of the analysis are further
summarized in Table S2 (cf. Section S7 of Supplement).
3.2.3 Statistical correlations

Scatter plots of rebound deviation with environmental variables of ambient temperature,

wind speed, and wind direction show no correlation for both daytime and nighttime datasets (cf.
Figure S1 of the Supplement). Scatter plots of rebound deviation with some possible
anthropogenic influences are presented in Figure S2. Soot, typically characterized by a solid core
region of black carbon, is expected in abundance both in urban pollution and biomass burning
emissions. There was, however, no correlation between rebound deviation and black carbon
concentrations. There was also no correlation between rebound deviation and total particle mass
concentration. Rebound deviation and sulfate concentration weakly anti-correlated, which might
be expected given the hygroscopicity of sulfate. Sulfate concentrations, however, were a poor



indicator of anthropogenic influence in this region because the variability in background
concentrations was comparable in magnitude to any urban influence(de Sá, 2016). As a caveat,
an assumption in correlation tests is that variance arises from a single variable, and the
possibility of two or more contributing or interacting factors is not directly considered.
Given that water uptake is an important process for softening organic material, scatter
plots of the rebound deviation at 75% RH with the hygroscopicity parameter κ are shown in
Figures 7a and 7b for the data sets from IOP1 and IOP2, respectively. Lower values of κ
represent decreased equilibrium water uptake for a fixed RH (Petters and Kreidenweis, 2007).
The plots show that the rebound deviation increased as the κ value decreased, meaning that
particles of lower hygroscopicity were less prone to being in liquid form. The apparent shift to
higher κ values from IOP1 to IOP2 arose from instrumental methods. During IOP1, κ values
were measured using the impactor apparatus at sub-saturated RH (i.e., < 100%) (cf. Supplement
S8). During IOP2, κ values were measured using a size-resolved CCN instrument at super-
saturated RH (i.e., > 100%) (Thalman, 2016). In principle, the two kinds of κ values should have
similar numerical values for water-soluble species (Petters and Kreidenweis, 2007). The
observed shift is consistent with a more frequent occurrence of non-liquid particles during IOP2
(Pajunoja et al., 2015), in this case as a result of more extensive biomass burning, as well as the
possibility of liquid-liquid phase separation (Renbaum-Wolff et al., 2016).
3.2.4 Chemical characteristics of rebounded particles
An analysis of the relationship between rebound deviation and chemical characteristics is
presented in Figure 8 based on the fractional loading of PMF group B. The data sets are
segregated for presentation by IOP1/IOP2, daytime/nighttime, and four bands of fractional
loading. Within each panel, box-whisker statistics of rebound deviation are shown for different



windows of relative humidity, ranging from 50% to 95%. The figure shows that the rebound
deviation increased at all RH values as the fractional loading of group B increased. The
background-average curve used as the reference for rebound deviation corresponded to a
fractional loading of 0.00 to 0.15 for group B or correspondingly of 0.85 to 1.00 for group A. An
increasing fractional loading of group B represented greater anthropogenic influence. The
inference is that anthropogenic influences, represented by a combination of urban pollution and
biomass burning, affected chemical composition in ways that increased the presence of non-
liquid PM above 50% RH.

Scatter plots between rebound deviation at 75% RH and the fractional loading of group B

are shown in Figures 9a and 9b for the data sets of the two IOPs. The data points are colored
according to the value of the hygroscopicity parameter κ. The plots show that rebound deviation
increased for low hygroscopicity and high fractional loading of group B, in agreement with the
presentation in Figures 7a and 7b and Figure 8. Figures 9a and 9b further show in explicit
fashion that the highest rebound deviations occurred during time periods affected by biomass
burning and urban pollution, as characterized by the lowest values of κ and the highest fractional
loadings of group B. Smaller κ values for lower fractional loadings of group B can be explained
by the differences in O:C ratios between group A and B (Massoli et al., 2010): the O:C ratios
were 0.95/0.95 (IOP1/IOP2) and 0.42/0.54 for groups A and B, respectively.

A model to predict rebound deviation based on chemical characteristics was constructed.

The fractional loadings of PMF group A and B were used as model inputs, and model
coefficients represented the effects of RH across nine bands (cf. Section S9 of the Supplement).
The observed and predicted rebound deviations are plotted in Figure 10. The corresponding
coefficients $R^2$ of determination were 0.65 and 0.72 for IOP1 and IOP2, respectively. Predicted



values were biased high for low rebound deviation and biased low for high rebound deviation.
The magnitudes of the model coefficients represented the relative importance of the two PMF
groups in predicting rebound deviation (Table S4). In this regard group B dominated during both
IOPs. This result is consistent with the role of anthropogenic influences in shifting the PM
population to fewer liquid particles and more non-liquid particles.

Analysis by STXM/NEXAFS supports the foregoing narrative of anthropogenic

influence as a modulator between liquid and non-liquid PM. Rebounded particles were collected
on October 1 during a time period classified as influenced by biomass burning emissions. The
carbon K-edge spectrum is shown in Figure 11a, and the STXM image is shown in Figure 11b. A
notable feature of the NEXAFS spectrum of the rebounded particles is the strong double bond.
Pöhlker et al. (2012) previously collected NEXAFS spectra for samples collected at a
background site in central Amazonia, and the strong feature of a double bond was absent. The
spectra instead resembled those of different types of reference biogenic secondary organic
material.

For comparison to the spectrum collected of the rebounded particles, carbon K-edge

spectra are shown for carbonaceous particles collected in other field and laboratory studies.
Based on these results as well as those of the rebound measurements, a hypothesis of soot or
black carbon to explain the rebounding particles was ruled out on three grounds. The double-
bond feature was homogeneously distributed throughout the particles (Figure 11b) compared to
inclusions that are typical for soot (Moffet et al., 2013;Knopf et al., 2014;O'Brien et al., 2014),
rebound deviation and black carbon concentrations did not correlate (Figure S2), and the
spectroscopic signatures of rebounded particles and soot did not match (Figure 11a). A
hypothesis of VOC-derived secondary organic material, including possible changes because of



shifts from $HO_2$ to NO-dominant chemistry (Liu et al., 2016a;Liu et al., 2016b), was also
eliminated by comparison of the NEXAFS spectrum of the rebounded particles to the reference
spectra for laboratory samples. An important caveat to these interpretations is that some of the
hypotheses ruled out for this particular day might still have a role to play on other days.

There was exceptional uniformity in the particle population characterized by

STXM/NEXAFS, which could suggest that the rebounding PM represented distant sources or
alternatively a strong single nearby source. The gray region around the red line in Figure 11a
illustrates the low variability across the population of analyzed particles (Figure 11b). Moreover,
the variability in the O:C ratio determined by the NEXAFS analysis was just ± 0.01 for O:C =
0.34 (± 0.03) [± 0.01], where the value in parentheses was the uncertainty of the measurement
and the value in the bracket was the variability across the image in Figure 11b. During the same
time period, the O:C ratio of the ambient PM was 0.77 ± 0.04 by AMS measurements. Hence,
the rebounding particles were significantly less oxidized, as is consistent with lower
hygroscopicity, increased double bond (C=C) content, and increased fractional loading of PMF
group B.

Several speculations can be made for the origins of the particles leading to the data set of

October 1. The chemical constituents giving rise to the double bonds might derive from
biological degradation products from incomplete combustion, such as in biomass burning
(Tivanski et al., 2007;Keiluweit et al., 2010). Unexplained by this speculation, however, is the
absence of a potassium signature in the NEXAFS spectra, which is typical of most biomass
burning. Future collection of NEXAFS spectra would be well motivated for the several different
types of biomass burning in an Amazonian context, such as from nearby fields, regionally around
Manaus, two or three days away from other regions of South America, and up to a week away



from Africa. An alternative speculation for this data set is that solid organic particles produced
by the impact of raindrops on wet soil surfaces could be making a contribution to the rebounded
PM analyzed here (Joung and Buie, 2015). Wang et al. (2016) recently reported detection of the
airborne soil organic particles generated by this mechanism over agricultural fields in the central
plains of the USA, and the corridor from Manaus to T3 has many agricultural fields. The
rebounded particles collected at T3 and the agricultural particles reported in Wang et al. (2016)
both had a homogeneous distribution of double bonds and similar elemental ratios and
absorption features. Even so, preliminary analysis across the extended data set at T3 between
rebound and nearby precipitation did not show a clear correlation. Another speculation relates to
the importance of aromatic compounds as hardening agents. Several gas-phase aromatic
compounds, laden with double bonds, were measured during IOP1 and IOP2, including toluene,
benzene, trimethylbenzene, and xylenes by proton-transfer mass spectrometry (Liu et al., 2016b).
Rebound deviation correlated positively with the concentrations of these compounds during both
IOPs, and the correlation was stronger during the night. Laboratory studies show that the uptake
of polycyclic aromatic hydrocarbons during the formation of biogenic PM can increase the
viscosity of the PM (Vaden et al., 2011;Zelenyuk et al., 2012;Abramson et al., 2013;Liu et al.,

2015).

**4. Conclusions**

Under background conditions, particles composed primarily of highly-oxidized biogenic

PM were hygroscopic, and they were liquid for the RH values prevailing over Amazonia at
surface level. Anthropogenic influences of urban pollution and biomass burning decreased
hygroscopicity, and non-liquid PM became more favored. The shift in physical state correlated
with decreasing values of the hygroscopicity parameter $\kappa$, decreasing O:C elemental ratios,



increasing concentrations of C=C functionalities, and increasing fractional loadings of AMS
PMF group B, all of which were indicative of anthropogenic influences. These results
demonstrate the importance of anthropogenic influences for altering the physical properties of
ambient particulate matter over tropical forests.

**Acknowledgments**. Institutional support was provided by the Central Office of the Large Scale

Biosphere Atmosphere Experiment in Amazonia (LBA), the National Institute of Amazonian

Research (INPA), and Amazonas State University (UEA). The Office of Biological and

Environmental Research of the Office of Science of the United States Department of Energy is

acknowledged for funding, specifically the Atmospheric Radiation Measurement (ARM) Climate

Research Facility, the Atmospheric System Research (ASR) Program, the Division of Chemical

Sciences, Geosciences, and Biosciences (Advanced Light Source at Lawrence Berkeley National

Laboratory, Beamlines 5.3.2 and 11.0.2), the Environmental Molecular Sciences Laboratory

(EMSL), and Pacific Northwest National Laboratory (PNNL). Further funding was provided by

the Amazonas State Research Foundation (FAPEAM), the São Paulo State Research Foundation

(FAPESP), the Brazil Scientific Mobility Program (CsF/CAPES), the USA National Science

Foundation, and the Japanese Ministry of the Environment. The work was conducted under

scientific licenses 001030/2012-4 of the Brazilian National Council for Scientific and

Technological Development (CNPq).

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



**Figure Captions**

**Figure 1**.    Rebound fraction as a function of apparatus relative humidity during IOP1 (blue)

and IOP2 (red), corresponding respectively to the wet and dry seasons in central

Amazonia. Panel (a) shows all measurements. Panel (b) shows the subset of data for

which the apparatus RH matched the ambient relative humidity (cf. Supplement

S3). Panel (c) represents the probability density function of relative humidity at the

T3 site during the wet and dry seasons of 2014. Points represent rebound

measurements for particles having mobility diameters of 190 nm. Statistics of

measurements group by relative humidity are represented in box-whisker format.

The horizontal line within a box indicates the median of the points, the horizontal

lines at the box boundaries indicate quartiles, and horizontal lines most distant from

the box indicate 10% and 90% quantiles. For comparison, the black line shows the

rebound curve for particles of secondary organic material produced by

photooxidation of an isoprene/α-pinene mixture in the Harvard Environmental

Chamber (Bateman et al., 2015).

**Figure 2**.    Probability density function of rebound fraction during (a) IOP1 and (b) IOP2 for

categorization by type of air mass and time of day: background conditions (green),

Manaus pollution (red), and biomass burning (orange). Solid and dashed lines

respectively represent daytime (12:00 to 16:00, local time; 16:00 to 20:00, UTC)

and nighttime measurements (23:00 to 04:00, local time; 03:00 to 08:00 UTC).

Results are shown for an apparatus RH of 75% and for particles having a mobility

diameter of 190 nm.



**Figure 3.** Deviation in rebound fraction relative to the average curve for background conditions during (a) IOP1 and (b) IOP2. Red, purple, and green coloring correspond respectively to the RH regions of rebound, transition, and adhesion for the background-average curve. Results are shown for particles having a mobility diameter of 190 nm.

**Figure 4.** Time series of representative species during two pollution events for IOP1 and IOP2 (left and right sides of figures, respectively). "Total mass concentration" plotted in the figure represents the sum of the AMS and black carbon mass concentrations measured for submicron PM. The PMF mass fractions are expressed in relative terms to one another and necessarily sum to unity. Time on the abscissa is expressed in UTC.

**Figure 5.** Representative TEM images of (a) adhering and (b) rebounding PM at 95% RH collected between 14:15 and 18:15 (UTC) on September 30, 2014, during IOP2. The rebound fraction during the time period of collection approached 0.3. The arrows in panel (b) highlight the locations of particles having high vertical dimensions and spherical or dome-like morphologies, as expected for solid particles.

**Figure 6.** Deviation in rebound fraction for categorization by type of air mass and time of day. The box-whisker representation of the 10%, 25%, 50%, 75%, and 90% quantiles of statistics for each RH bin is explained in the caption to Figure 1. Air mass categorization is as for Figure 2. Results are shown for particles having a mobility diameter of 190 nm.





**Figure 7.** Scatter plot of rebound deviation with the hygroscopicity parameter κ for (a) IOP1 and (b) IOP2. For clarity of presentation, data points are shown for apparatus RH values between 73 and 78%, although the trend applies more broadly. As a guide to the eye, in each panel data are divided into four groups, and medians and quartiles of the groups are plotted as black circles and whiskers, respectively.

**Figure 8.** Deviation in rebound fraction for categorization by chemical characteristics. The box-whisker representation of the 10%, 25%, 50%, 75%, and 90% quantiles of rebound fraction is explained in the caption to Figure 1. The chemical characteristics are categorized by the fractional loading of PMF group B as < 0.15 (green), 0.15 to 0.3 (blue), 0.3 to 0.6 (orange), and > 0.6 (red).

**Figure 9.** Scatter plot of rebound deviation with the fractional loading of PMF group B for (a) IOP1 and (b) IOP2. The data points are colored according to the corresponding hygroscopicity parameter κ. For clarity of presentation, data points are shown for apparatus RH values between 73 and 78%, although the trend applies more broadly. As a guide to the eye, in each panel data are divided into four groups, and medians and quartiles of the groups are plotted as black circles and whiskers, respectively.

**Figure 10.** Scatter plot of observed compared to predicted rebound deviation for (a) IOP1 and (b) IOP2. Predictions are based on linear combinations of the loadings of the two PMF groups. The linear coefficients used in the prediction were optimized as a function of RH (Table S4). The solid line represents a one-to-one correlation, and the dashed line represents the best linear fit. Coefficients $R^2$ of determination were 0.65 and 0.72 for the IOP1 and IOP2 datasets, respectively. Points are color-coded by relative humidity.



**Figure 11.** STXM/NEXAFS analysis of particles collected after rebound from the impaction plate. Samples were collected between 1:00 and 10:00 (UTC) of October 1, 2014. (a) Carbon K-edge spectrum of rebounded particles (red). Lines at 285.4 and 288.5 eV highlight absorption by double bonds (C=C) and carboxylic acids (-COOH), respectively. The line in the region of 286.5 to 286.7 eV can have contributions from ketones, carbonyl-substituted aromatics, and phenolic species. For comparison, spectra are shown for carbonaceous particles collected in other field and laboratory studies: soil organic particles from central USA (Wang et al., 2016), soot from the ambient environment, atmospheric particles collected at a background site in central Amazonia (Pohlker et al., 2012), and three laboratory samples of secondary organic material. Data sources: isoprene-derived SOM produced under $HO_2$-dominant conditions (O'Brien et al., 2014); isoprene-derived SOM produced under NO-dominant conditions (O'Brien et al., 2014), and toluene-derived SOM produced under NO-dominant conditions (this study). (b) STXM image from which the NEXAFS spectrum of panel (a) was obtained. Coloring is by red for absorption at 285.4 eV (i.e., double bonds). Coloring is by green for other types of functionalities. The pixelation visible in the image corresponds to the STXM spatial resolution during data collection.



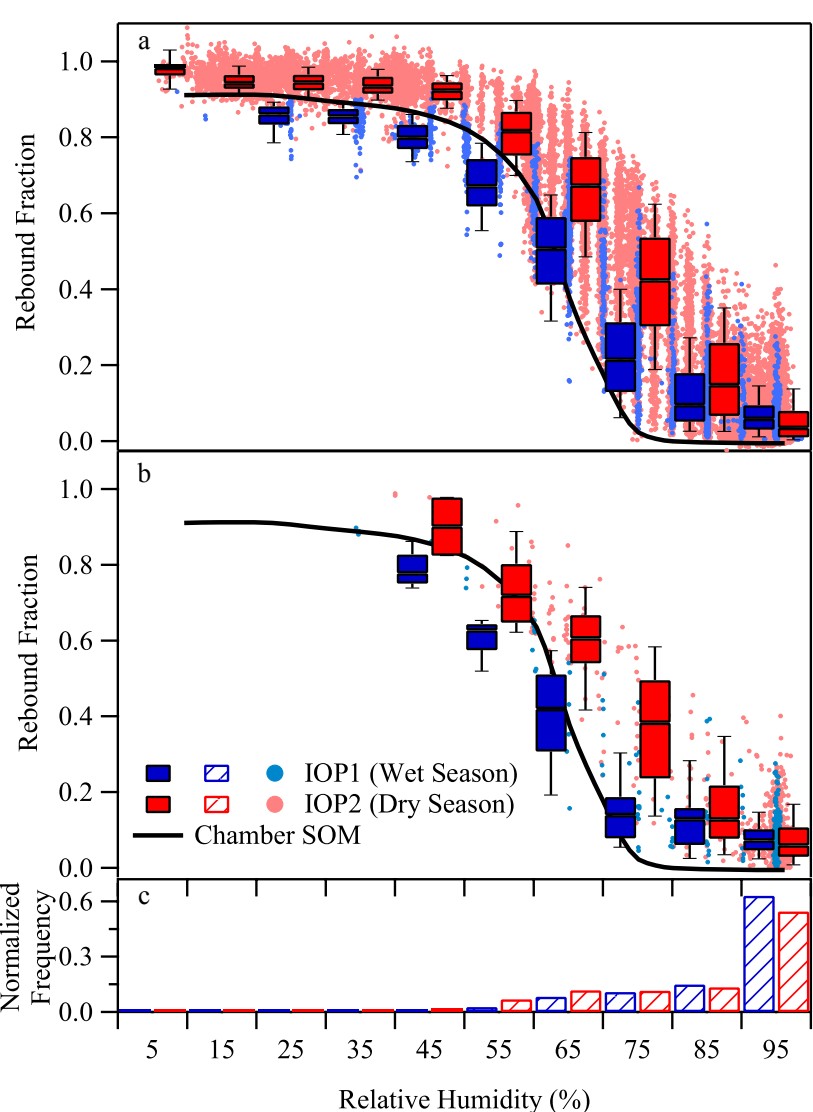

Figure 1



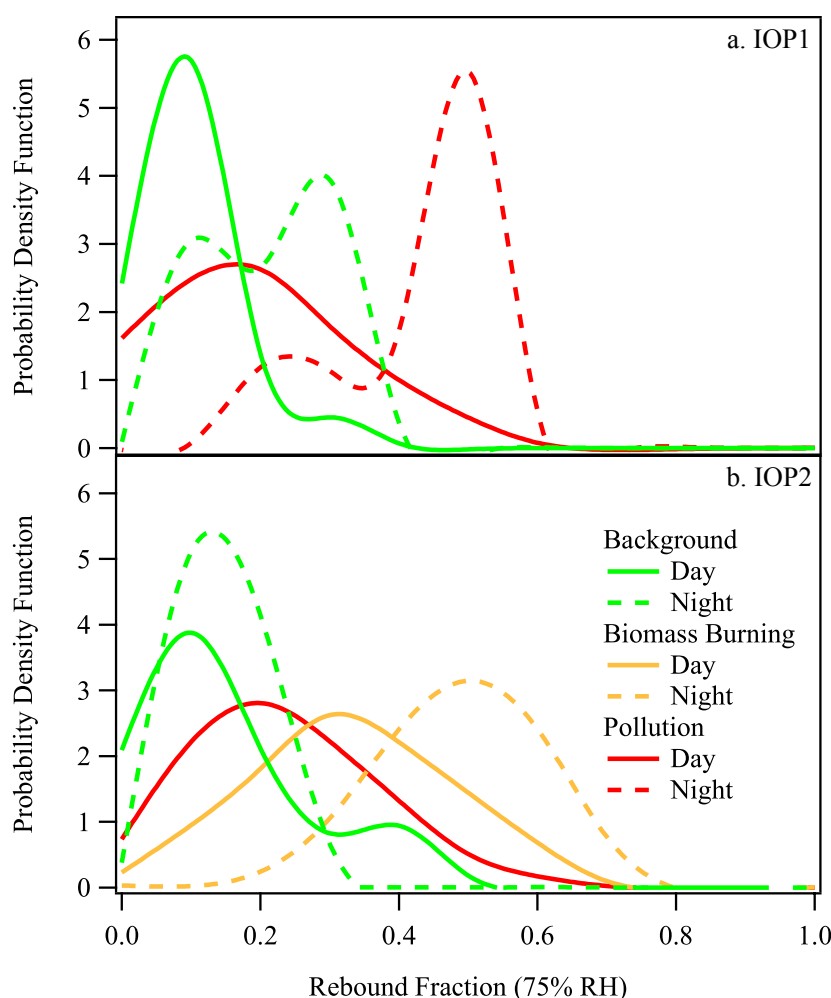

Figure 2





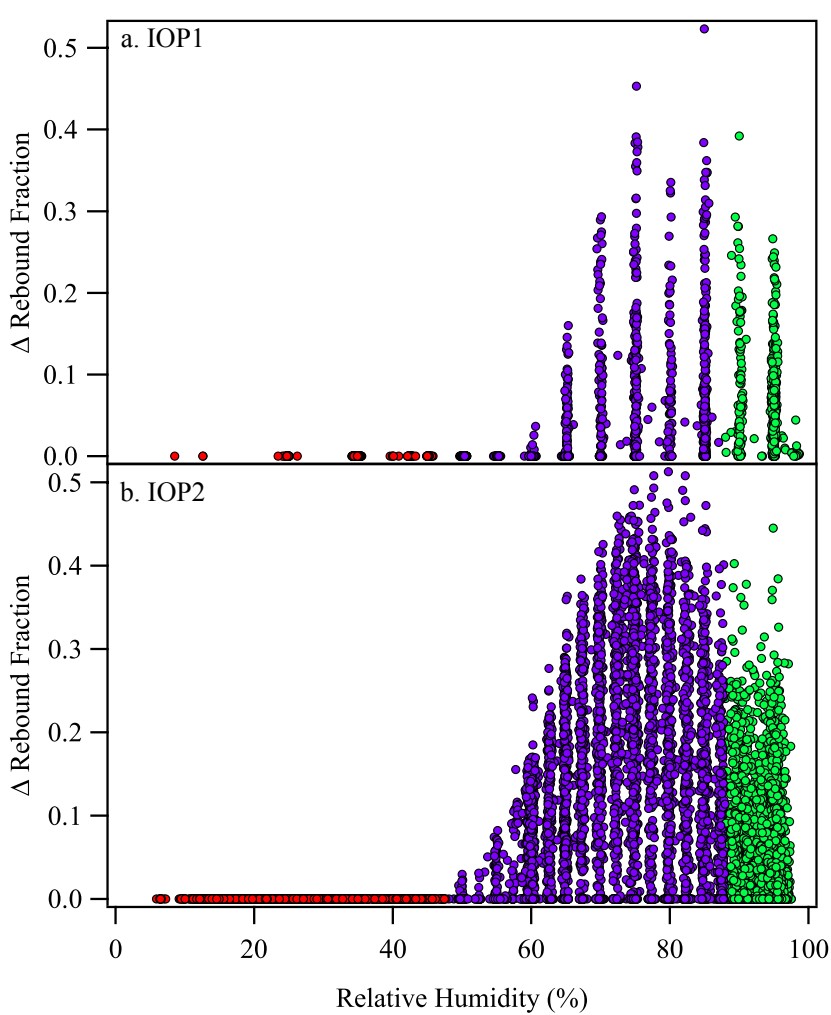

Figure 3



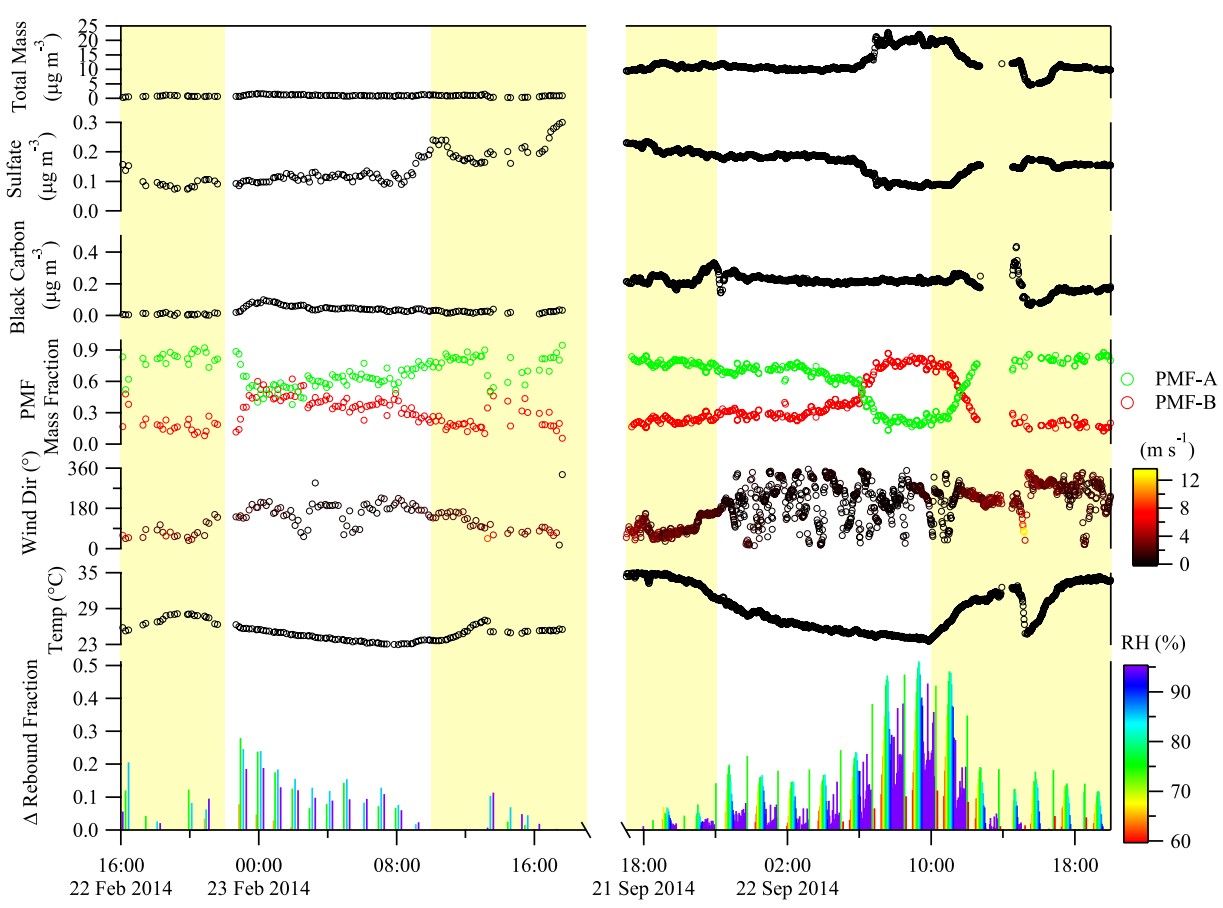

Figure 4





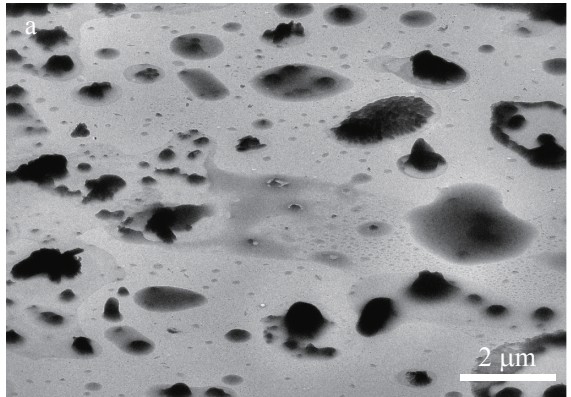 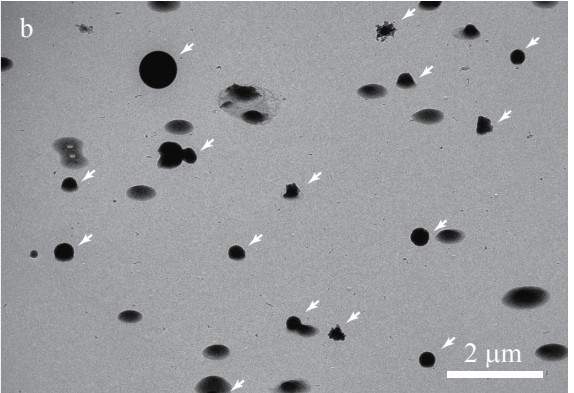

Figure 5





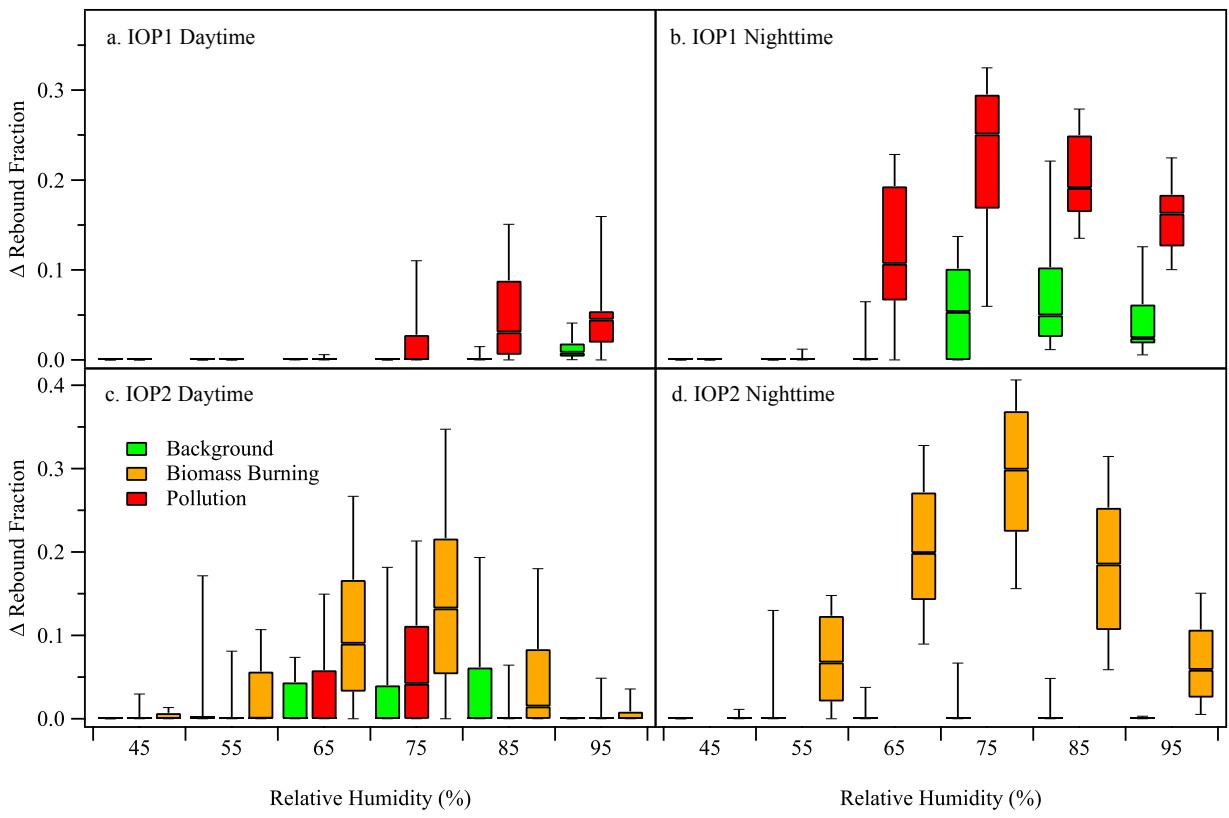

Figure 6





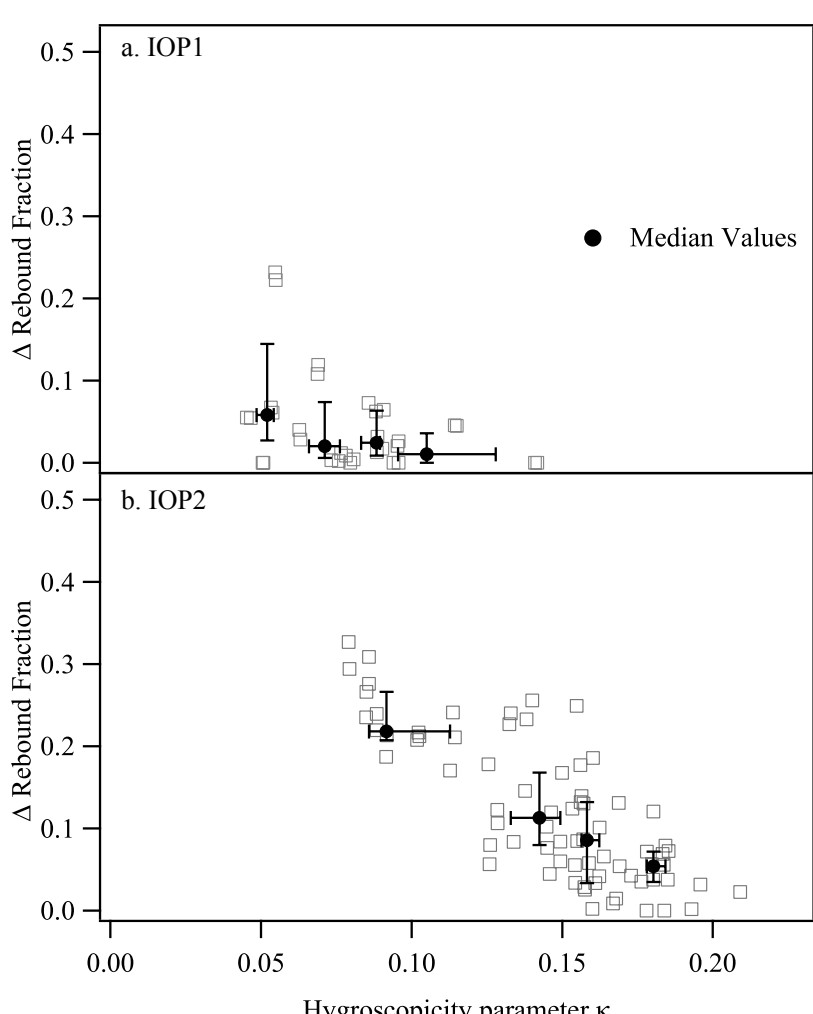

Figure 7

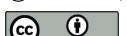



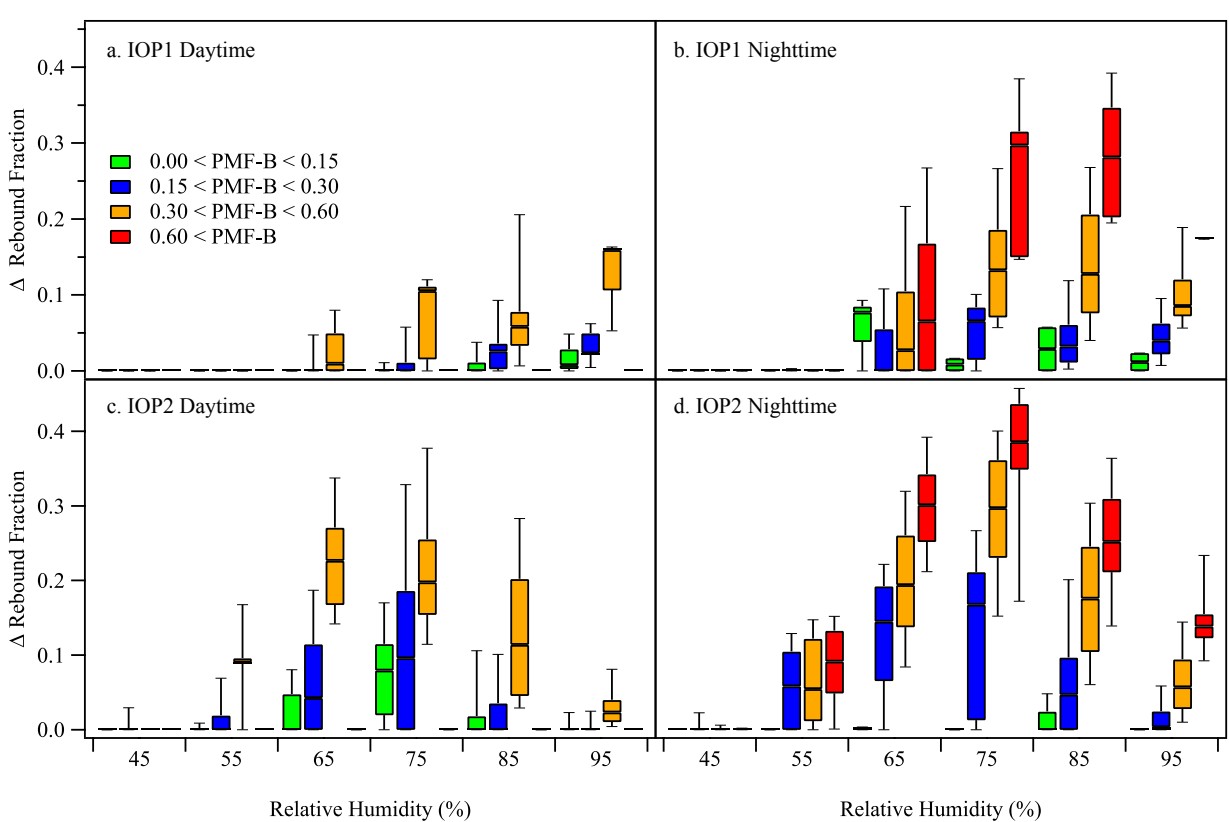

Figure 8





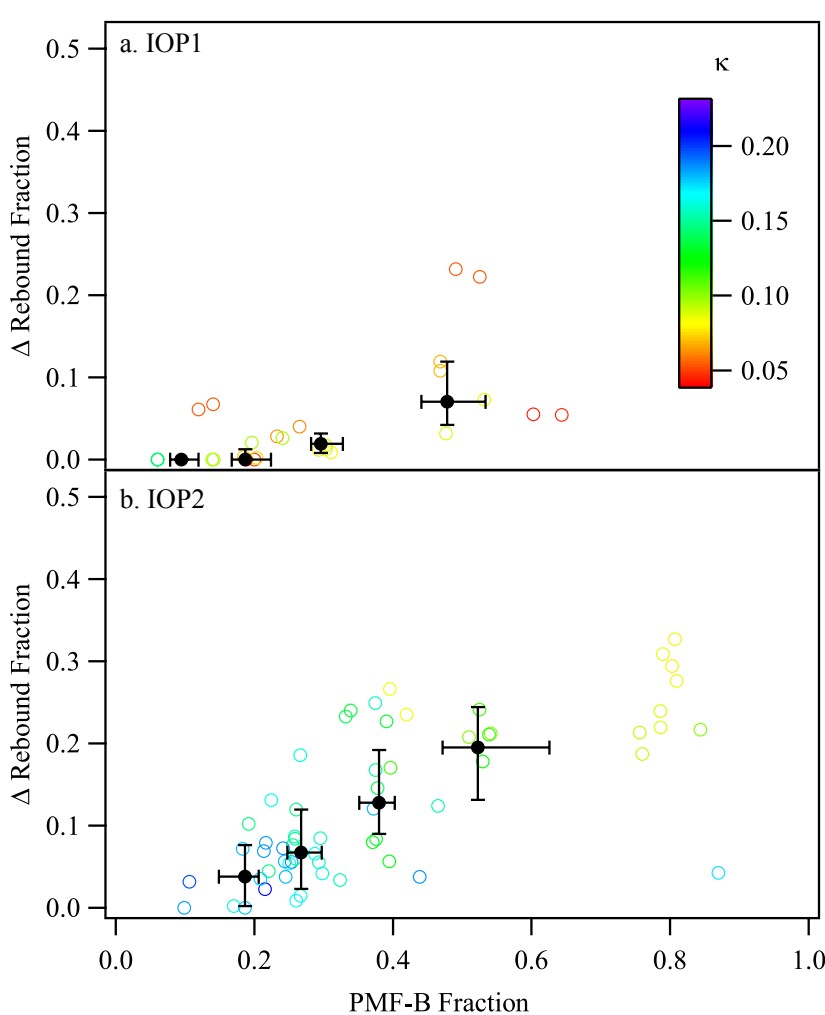

Figure 9







Figure 10





Figure 11