# Peer review of "Anthropogenic influences on the physical state of submicron particulate"

_Atmospheric Chemistry and Physics, 2016_

## Referee Comment (RC1) · Anonymous Referee #1 · 19 Sep 2016

**General comments:**

The paper by Bateman describes a results from a field campaign in the Amazon, focusing on the physical state of aerosol particles with an experimental method developed recently by the same group. In particular, particle rebound fraction within an special-purpose impactor setup is used as an indicator of physical state, with the ability to distinguish between liquid particles (i.e., low rebound fraction) and particles of a semi-solid or solid phase state (i.e., medium to high rebound fraction). This method is then used to measure rebound fraction as a function of relative humidity (varied within the impactor; impRH), and ambient parameters such as temperature, humidity, particle O:C ratio, and the concentration of varies gases. It is shown that the particles'

physical state is high at low impRH and low at high impRH, in agreement with previous measurements and with current understanding of the dependence of physical state and of viscosity upon humidity.

What is new in this study is that the particles' physical state varied for periods when the air was influenced by anthropogenic activities such as biomass burning or air originating from the city of Manaus. The higher rebound fraction and, thus, the larger fraction of semi-solid and/or solid particles is in agreement with particle chemistry: the particles are less oxidized and of a lower hygroscopicity than biogenic particles, with a significant contents of C=C double bonds indicative of aromatic VOCs arising from anthropogenic sources. The higher rebound fraction is also in agreement with electron microscopy images showing rather flat particles under conditions of low rebound fraction (and thus impaction of rather low-viscosity liquid) and particles with more pronounced vertical extension und conditions of higher rebound fraction (and thus particles whose shape was likely not influenced during impaction, indicative of more solid-like particles). The authors then utilized aerosol mass spectra as well as various other tracers and parameters measured in parallel to develop a positive-matrix factorization (PMF) analysis that was able to explain the observed rebound fraction to a significant degree.

The manuscript presents a significant contribution to atmospheric aerosol science in the Amazon region and the influence of anthropogenic activities upon the aerosols' physical state. In general, the manuscript was a pleasure to read and the manuscript text and the figures were prepared thoroughly. Moreover, also the measurements appear to have been conducted with great care. In my opinion the manuscript can be published more or less as is, and I have only very minor suggestions that should be taken care of in a revised version.

**Minor and technical comments:**

(1) lines 105-107: '. . . the anthropogenic molecules have a tendency to reduce water uptake and thereby REDUCE the viscosity of the mixed particles.' I believe you meant to say '. . . ENHANCE the viscosity'.

(2) lines 101-103: these statements are also supported by the model calculations of Berkemeier et al. (Atmos. Chem. Phys., 14, 12513–12531, 2014.) suggesting that the anthropogenic aromatic SOA precursors naphthalene may lead to higher-viscosity secondary organic material when compared to biogenic precursors such as pinene and isoprene.

(3) In Figure 2b the dashed red line indicating 'pollution at night' is missing. Is this by accident or intentional?

(4) lines 368-379 and Figure 7: the hygroscopicity was measured using different approaches. This is openly discussed in the text, but I would like you to mention that $\kappa_{CCN}$ and $\kappa_{HGF}$ can be quite different for solutes that form non-ideal aqueous solutions. Therefore, I suggest to indicate in the Figure (or at least in the figure caption) that the plotted $\kappa$-values were obtained by different methods at different humidity.
* * *

---

## Referee Comment (RC2) · Anonymous Referee #2 · 8 Nov 2016

The paper reports particle bounce measurements made in Amazonia during GOAmazon2014 campaign. According to the results, particles are mostly liquid like in Amazonia and the anthropogenic influence can be seen in increased fraction of bounced particles. The paper is well written and the results are interesting enough so that I can recommend that the paper could be accepted to ACP after following comments have been addressed by the authors.

1) Page 11, line 226: "Calibration of the impactor shows a transition from rebound to adhesion between 102 to 1 Pa s in viscosity for sucrose particles (Bateman et al., 2015)." To relate particle bounce to viscosity values, more than one substance should be used in calibrations. The particle bounce properties are not affected only by viscosity, but

also other material characteristics which may vary between different substances.

2) Page 11-12, lines 245-246: It seems that authors interpret the bounce curves in a way that if the rebound fraction is 5%, approximately 5% of the particles are solid. Even the bounce curves for single components, such as sucrose, are s-shaped curves and rebound value varies between 1 and 0. In the case of sucrose particles all the particles have absorbed same amount of water at certain RH, hence they all have the same physical phase state. The reason why for example 20% of sucrose particles are bouncing off from the impactor is not because 20% of the particles are solid and 80% liquid. 20% of sucrose particles are bouncing off at certain RH and 80% are sticking on the impactor substrate because the velocity of particles in impactor jet depends on their radial distance from the center of the impactor jet. Hence in this example, 20% of sucrose particles have kinetic energy higher than the dissipation and surface adhesion energies. So it is obvious that increasing rebound fraction doesn't necessarily imply external mixture and increasing fraction of solid particles. It can also imply changes in material characteristics of all particles (more solid particles). This should be taken into account in data interpretation and also the text should be changed accordingly throughout the manuscript.

3) According to main conclusions of the paper the changes in measured rebound correlated with decreasing kappa and decreasing O:C. Still no data on O:C is shown in the whole manuscript. Authors should show, for example, how O:C varies between pollution, biomass burning and background cases and also during day and night (Figure 3). Also O:C panel should be added to figure 4.

4) Also the rebound data corresponding different cases (polluted-biomass burning-background) should be shown.

---

## Author Comment (AC1) · 19 Dec 2016

We thank the reviewers for the input and resulting improvements to the manuscript.

**Response to Reviewer 1:**

**Minor and technical comments:**
(1) lines 105-107: '. . . the anthropogenic molecules have a tendency to reduce water uptake and thereby REDUCE the viscosity of the mixed particles.' I believe you meant to say '. . . ENHANCE the viscosity'.

Thank you. The correction is made.

(2) lines 101-103: these statements are also supported by the model calculations of Berkemeier et al. (Atmos. Chem. Phys., 14, 12513–12531, 2014.) suggesting that the anthropogenic aromatic SOA precursors naphthalene may lead to higher-viscosity secondary organic material when compared to biogenic precursors such as pinene and isoprene.

This reference is added.

(3) In Figure 2b the dashed red line indicating 'pollution at night' is missing. Is this by accident or intentional?

For Figure 2b, there were no data that fit the classification of "pollution at night". The figure caption is clarified, as follows: *"No data sets fit the classification of Manaus pollution during the nighttime of IOP2 (i.e., absence of red dashed line in panel (b))."*

(4) lines 368-379 and Figure 7: the hygroscopicity was measured using different approaches. This is openly discussed in the text, but I would like you to mention that _CCN and _HGF can be quite different for solutes that form non-ideal aqueous solutions. Therefore, I suggest to indicate in the Figure (or at least in the figure caption) that the plotted _-values were obtained by different methods at different humidity.

$\kappa_{CCN}$ and $\kappa_G$ are now included in Figure 7. The following clarification is added to the figure caption: "*Different techniques were used to measure $\kappa_G$ and $\kappa_{CCN}$, as described in the main text.*"

The main text is changed by deleting two sentences and replacing them with the following: "Sub- and supersatured $\kappa$ values can be systematically different (Petters and Kreidenweis, 2007; Ruehl et al., 2016)."

---

## Author Comment (AC2) · 19 Dec 2016

We thank the reviewers for the input and resulting improvements to the manuscript.

**Response to Reviewer 2:**
1) Page 11, line 226: "Calibration of the impactor shows a transition from rebound to adhesion between 102 to 1 Pa s in viscosity for sucrose particles (Bateman et al., 2015)." To relate particle bounce to viscosity values, more than one substance should be used in calibrations. The particle bounce properties are not affected only by viscosity, but also other material characteristics which may vary between different substances.

We agree with the reviewer that a more robust understanding of material physical properties as related to their rebound behavior would be beneficial for a more complex analysis of ambient particle rebound behavior. The transition from semisolid to liquid for a variety of inorganic compounds with respect to rebound behavior in this same apparatus has recently been studied (Li et al., 2016). In the absence of a complete study on a variety of material characteristics and their rebound response, the present study assumes that the uptake of water by semi-solid sucrose particles to liquid is a valid model system for the transition of SOM from semi-solid to liquid.

The text is expanded as follows: "*As a point of reference, the transition from rebound to adhesion occurs across a viscosity transition of $10^2$ to 1 Pa s for sucrose particles for the operating conditions of the impactor (Bateman et al., 2015). The viscosity range corresponding to the rebound transition can depend on particle composition, but this aspect is not investigated herein.*"

2) Page 11-12, lines 245-246: It seems that authors interpret the bounce curves in a way that if the rebound fraction is 5%, approximately 5% of the particles are solid. Even the bounce curves for single components, such as sucrose, are s-shaped curves and rebound value varies between 1 and 0. In the case of sucrose particles all the particles have absorbed same amount of water at certain RH, hence they all have the same physical phase state. The reason why for example 20% of sucrose particles are bouncing off from the impactor is not because 20% of the particles are solid and 80% liquid. 20% of sucrose particles are bouncing off at certain RH and 80% are sticking on the impactor substrate because the velocity of particles in impactor jet depends on their radial distance from the center of the impactor jet. Hence in this example, 20% of sucrose particles have kinetic energy higher than the dissipation and surface adhesion energies. So it is obvious that increasing rebound fraction doesn't necessarily imply external mixture and increasing fraction of solid particles. It can also imply changes in material characteristics of all particles (more solid particles). This should be taken into account in data interpretation and also the text should be changed accordingly throughout the manuscript.

Thank you for this thoughtful comment. It is true that there is variation in the impact velocity and thus rebound fraction as a function of radial distance from the center of the impactor jet. We have modeled the flow-dynamics of our impactor and for the nozzles employed in this study (cf. Figure S3 and S4 of Bateman et al., 2014) and found the kinetic energy of the impacting particles at large diameters should be well above the adhesion energy even at high RH (cf. Figure 7 of the Bateman et al., 2014). This is the main reason for using particle diameters of 190 nm. Under the controlled conditions and thorough understanding of the particle flow dynamics of our impactor we can rule out any change in rebound fraction attributed to kinetic energy and surface adhesion.

We also agree that the reviewer is correct that we assume the particles are a homogeneous mixture during most of the analysis. To this point the caveat is included: *"...that increase viscosity when internally mixed with background PM and increased concentrations of non-liquid anthropogenic particles in external mixtures of anthropogenic and biogenic PM."* Overall, to the reviwer's point, the width of the rebound curve exceeds the width of calibration particles, indicating that there are different particle types undergoing nonliquid/liquid transitions at different RH values.

3) According to main conclusions of the paper the changes in measured rebound correlated with decreasing kappa and decreasing O:C. Still no data on O:C is shown in the whole manuscript. Authors should show, for example, how O:C varies between pollution, biomass burning and background cases and also during day and night (Figure 3). Also O:C panel should be added to figure 4.

The intention was to use PMF-A and PMF-B as the main supporting data for the changes in the measured rebound fractions. To avoid the confusion highlighted by the reviwer's comment, we removed the statement *"...decreasing O:C elemental ratios,"* from the conclusions in the manuscript.

We addressed the reviewers concern that there is not enough data included for O:C ratios by adding Table S3 to the supporting information. This table includes the average O:C ratios for the various air mass classification categories and day/night time periods used in the manuscript. Text is added as follows: "*The average O:C ratios for the various air-mass classifications can be found in Table S3 of the Supplement."* The O:C panel is added to Figure 4.

4) Also the rebound data corresponding different cases (polluted-biomass burning background) should be shown.

These data sets are shown in Figure S1 as whisker-box plots. Test is added as follows, *"Figure S1 of the Supplement presents an additional level of detail."*